# Functional evidence implicating chromosome 7q22 haploinsufficiency in myelodysplastic syndrome pathogenesis

Jasmine C Wong[1], Kelley M Weinfurtner[1], Maria del pilar Alzamora[1], Scott C Kogan[2], Michael R Burgess[3†], Yan Zhang[4], Joy Nakitandwe[5], Jing Ma[5], Jinjun Cheng[5], Shann-Ching Chen[5‡], Theodore T Ho[6], Johanna Flach[6§], Damien Reynaud[6¶], Emmanuelle Passegué[6], James R Downing[5], Kevin Shannon[1*]

[1]Department of Pediatrics, University of California, San Francisco, San Francisco, United States; [2]Department of Laboratory Medicine, University of California, San Francisco, San Francisco, United States; [3]Division of Hematology/Oncology, University of California, San Francisco, San Francisco, United States; [4]Unit of Hematopoietic Stem Cell and Transgenic Animal Models, Institut Pasteur of Shanghai, Chinese Academy of Sciences, Shanghai, China; [5]Department of Pathology, St. Jude Children's Research Hospital, Memphis, United States; [6]Eli and Edythe Broad Center of Regeneration Medicine and Stem Cell Research, Department of Medicine, University of California, San Francisco, San Francisco, United States

**\*For correspondence:**
ShannonK@peds.ucsf.edu

**Present address:** †Celgene Corporation, San Francisco, United States; ‡Thermo Fisher Scientific, South San Francisco, United States; §Comprehensive Cancer Center, Institute of Experimental Cancer Research, Ulm, Germany; ¶Department of Pediatrics, Cincinnati Children's Hospital Medical Center, Cincinnati, United States

**Competing interests:** The authors declare that no competing interests exist.

**Abstract** Chromosome 7 deletions are highly prevalent in myelodysplastic syndrome (MDS) and likely contribute to aberrant growth through haploinsufficiency. We generated mice with a heterozygous germ line deletion of a 2-Mb interval of chromosome band *5A3* syntenic to a commonly deleted segment of human *7q22* and show that mutant hematopoietic cells exhibit cardinal features of MDS. Specifically, the long-term hematopoietic stem cell (HSC) compartment is expanded in *5A3*[+/del] mice, and the distribution of myeloid progenitors is altered. *5A3*[+/del] HSCs are defective for lymphoid repopulating potential and show a myeloid lineage output bias. These cell autonomous abnormalities are exacerbated by physiologic aging and upon serial transplantation. The *5A3* deletion partially rescues defective repopulation in *Gata2* mutant mice. *5A3*[+/del] hematopoietic cells exhibit decreased expression of oxidative phosphorylation genes, increased levels of reactive oxygen species, and perturbed oxygen consumption. These studies provide the first functional data linking 7q22 deletions to MDS pathogenesis.

## Introduction

The myelodysplastic syndromes (MDSs) are clonal stem cell disorders characterized by ineffective hematopoiesis, morphologic dysplasia, and a variable risk of progression to acute myeloid leukemia (AML) (*Elias et al., 2014*). Monosomy 7 (−7) and deletions affecting the long arm of chromosome 7 [del(7q)] are highly prevalent acquired cytogenetic abnormalities in de novo and in therapy-related MDS and AML (t-MDS/t-AML) (*Smith et al., 2003*). The proportion of −7/del(7q) cells is markedly increased in the hematopoietic stem cell (HSC) and progenitor compartments of MDS patients relative to T and B lymphocytes (*Kere et al., 1987b*; *Abrahamson et al., 1991*; *Bernell et al., 1996*; *Will et al., 2012*; *Elias et al., 2014*). Recent studies demonstrating quantitative changes in the frequencies of phenotypic primitive long-term HSCs, common myeloid progenitors (CMPs), and granulocyte-monocyte progenitors (GMPs) in MDS patients with −7/del(7q) further support diverse effects on hematopoiesis (*Will et al., 2012*; *Pang et al., 2013*).

**eLife digest** Stem cells in the bone marrow are essential for creating new blood cells. Myelodysplastic syndrome (MDS) is a common type of blood cancer in the elderly that occurs when blood cells fail to develop normally. Depending on which types of blood cells are affected, individuals with MDS may bleed more easily, feel weak and tired, or be unable to effectively fight off infections.

Animals and plants store their genetic information in the form of chromosomes. Humans have 23 pairs of chromosomes, with one copy inherited from the mother, and the other from the father. The bone marrow cells of many people with MDS delete a section from one of their copies of chromosome 7. As this section contains many different genes, it is difficult to fully understand which specific genes contribute to the development of MDS when one copy is lost.

Wong et al. have now genetically engineered mice to lack a section of one of their copies of chromosome 7 that is often missing in patients with MDS. Bone marrow cells from these mice exhibit many of the same abnormalities found in human MDS. For example, most of the immature blood stem cells expand, but these stem cells do not correctly specialize into mature blood cells—in particular, not enough immune cells are produced. The developing blood cells also have problems expressing several genes, including one that helps to protect the cells from damaging molecules called reactive oxygen species. These problems worsen as the mice age.

These mice provide the first evidence directly linking the missing section of chromosome 7 to abnormalities found in MDS patients. Future studies using the mice will advance our understanding of how the loss of this section of chromosome 7 interacts with other genes involved in MDS to alter the course of this disease and how it responds to treatment.

Multiple studies of MDS and AML specimens with interstitial deletions on 7q have implicated three putative commonly deleted segments (CDSs) at chromosome bands 7q22, 7q34, and 7q35-36 (*Figure 1A*) (*Kere et al., 1987a*; *Le Beau et al., 1996*; *Fischer et al., 1997*; *Liang et al., 1998*; *Tosi et al., 1999*; *Jerez et al., 2012*; *McNerney et al., 2013*; *Hosono et al., 2014*). Of these intervals, 7q22 is deleted most frequently in cases of MDS or AML with a del(7q) (*Le Beau et al., 1996*). Targeted sequencing of candidate myeloid tumor suppressor genes (TSGs) located within a 2.5 Mb 7q22 CDS delineated by *Le Beau et al. (1996)*, and recent comprehensive genomic analyses of clinical specimens strongly implicate a haploinsufficient role of 7q22 deletions in leukemogenesis (*Kere et al., 1989*; *Fischer et al., 1997*; *Liang et al., 1998*; *Tosi et al., 1999*; *Ebert, 2011*; *Jerez et al., 2012*; *McNerney et al., 2013*; *Hosono et al., 2014*). Consistent with this proposed mechanism, biallelic inactivation of any 7q gene is rare in MDS patients with −7/del(7q). For example, whole exome sequencing of 68 myeloid malignancies characterized by −7/del(7q) uncovered recurrent mutations in only *EZH2* (located at 7q36; n = 4) and *LUC7L2* (at 7q34; n = 3). Biallelic inactivation of these genes and of *CUX1* (located at 7q22) was observed in a small cohort of patients with 7q isodisomy (*Hosono et al., 2014*). *SAMD9L*, *RASA4*, *DOCK4*, and *MLL3* are other 7q genes that have been implicated as contributing to leukemogenesis by haploinsufficiency or epigenetic transcriptional repression (*Figure 1A*) (*Asou et al., 2009*; *Zhou et al., 2011*; *Nagamachi et al., 2013*; *Chen et al., 2014*; *Poetsch et al., 2014*). Here, we demonstrate hematopoietic abnormalities in mice with a germ line deletion of a contiguous CDS of chromosome band 5A3 (*5A3$^{+/del}$*) syntenic to a 2.5 Mb 7q22 CDS delineated by *Le Beau et al. (1996)* that support a mechanistic role of 7q22 deletions in MDS pathogenesis.

## Results and discussion

### Abnormal differentiation and repopulation of 5A3$^{+/del}$ stem and progenitor cells

We generated mice carrying a 2 Mb germ line *5A3* deletion that removes 13 genes syntenic to a human 7q22 CDS (*Figure 1A*) (*Wong et al., 2010*). *5A3$^{+/del}$* mice are smaller than wild-type (WT) littermates, and homozygous deletion of the *5A3* region causes embryonic lethality before 10.5 dpc (data not shown). Total nucleated bone marrow (BM) cell counts as well as spleen and thymus weights are reduced in mutant animals (*Figure 1B–D*), which maintain normal peripheral blood cell counts.

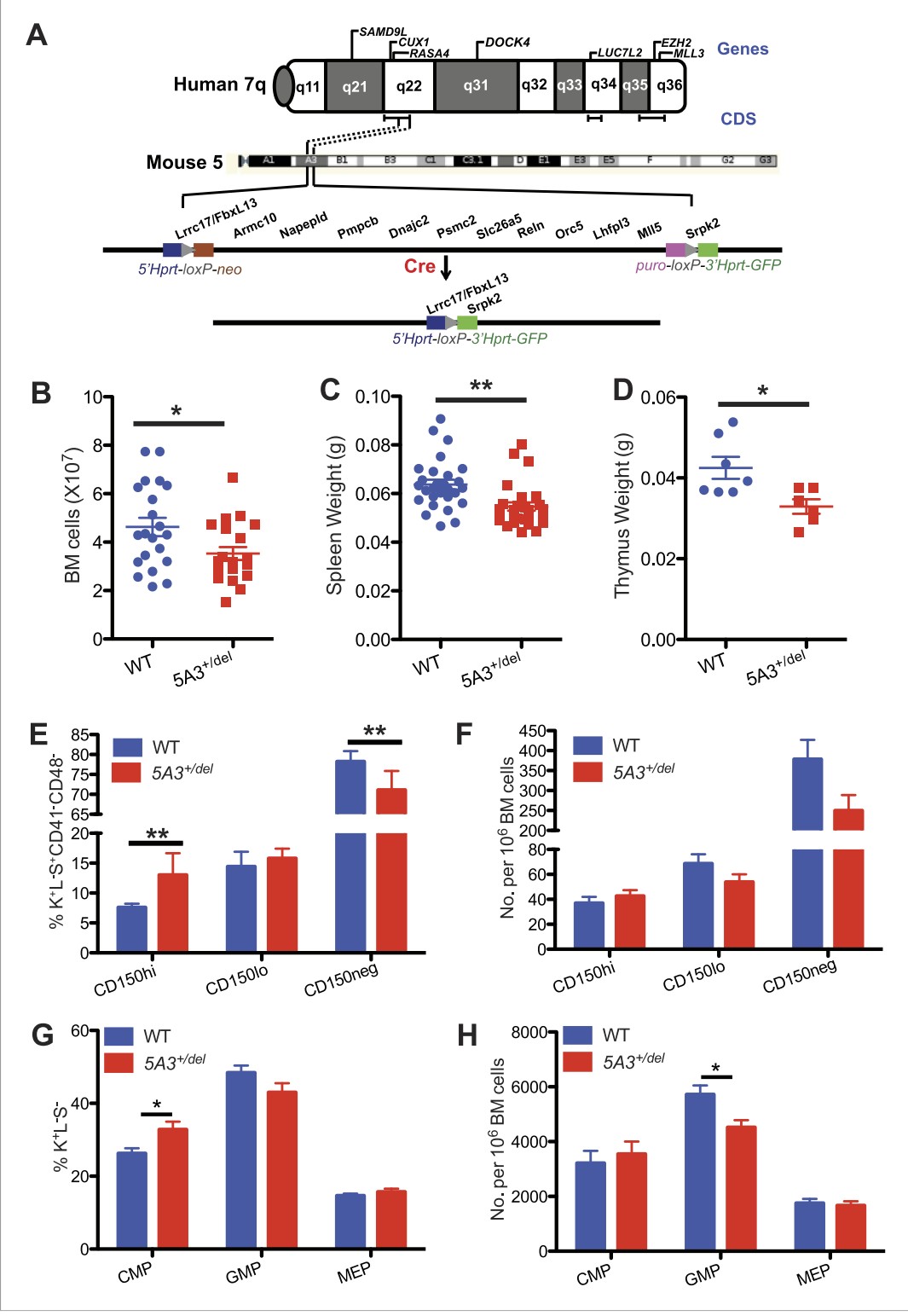

**Figure 1**. A heterozygous *5A3* deletion corresponding to human *7q22* perturbs steady-state hematopoiesis. (**A**) Top, candidate 7q myeloid tumor suppressor genes described previously (*Asou et al., 2009*; *Ernst et al., 2010*; *Nikoloski et al., 2010*; *Zhou et al., 2011*; *McNerney et al., 2013*; *Chen et al., 2014*; *Hosono et al., 2014*; *Poetsch et al., 2014*) appear above the diagram of chromosome 7q while commonly deleted segments (CDSs) within 7q22, 7q34, and 7q35-36 identified by different research groups (*Kere et al., 1987a*; *Le Beau et al., 1996*; *Fischer et al., 1997*; *Liang et al., 1998*; *Tosi et al., 1999*; *Jerez et al., 2012*; *McNerney et al., 2013*; *Hosono et al., 2014*) are
*Figure 1. continued on next page*

*Figure 1. Continued*

depicted by brackets immediately below it. Middle, dotted lines demarcate the segments of mouse chromosome 5A3 corresponding to the human 7q22 CDS targeted in this study. Bottom, expressing Cre recombinase in embryonic stem (ES) cells doubly targeted with LoxP sequences within the *Fbxl13* and *Srpk2* genes excised a 2-Mb interval. Gene order is based on the Ensembl database and is not drawn to scale. (**B**) Total numbers of bone marrow (BM) cells from 2 femurs and 2 tibiae in *5A3+/del* mice and wild-type (WT) littermates at 8–12 weeks of age. (**C, D**) Spleen (**C**) and thymus (**D**) weights in *5A3+/del* mice and WT littermates at 8–12 weeks of age. (**E, F**) Percent contributions (**E**) and frequencies (**F**) of cells with high (CD150hi HSC), low (CD150lo HSC), and absent CD150 expression (CD150neg MPP) within the K+L−S+ CD41−CD48− compartment of WT and *5A3+/del* mice at 8–12 weeks of age (n = 6 of each genotype). (**G**) Percent contribution of common myeloid progenitor (CMP), granulocyte-monocyte progenitor (GMP), and megakaryocyte erythroid progenitors (MEP) within the Lin−Sca1+c-kit+ compartment of *5A3+/del* mice and WT littermates. (**H**) Frequencies of CMP, GMP, and MEP in WT and *5A3+/del* BM. The error bars indicate S.E.M. with significant differences between WT and *5A3+/del* mice designated by asterisks as follows: *p < 0.05, **p < 0.01.

The following figure supplement is available for figure 1:

**Figure supplement 1**. Gating strategy for hematopoietic stem and progenitor populations.

---

Differential expression of CD150 distinguishes HSC populations with different self-renewal, differentiation, and repopulating potentials. Specifically, HSCs, with a surface c-kit+, lineage−, Sca-1+ (KLS), and CD150hi (CD150hi HSC) immunophenotype possess potent self-renewal capacity, are predisposed to myeloid differentiation and expand upon aging (*Kiel et al., 2005*; *Beerman et al., 2010a*; *Hock, 2010*). Strikingly, the proportion of CD150hi HSCs is increased in *5A3+/del* BM with a corresponding decrease in the percentage of CD150 negative multi-potent progenitors (CD150neg MPP) (*Figure 1E*, *Figure 1—figure supplement 1A*). This results in a normal frequency of CD150hi HSCs in *5A3+/del* mice, despite an overall reduction in the size of the stem/progenitor compartment (*Figure 1F*). The proportion of CMPs is elevated in *5A3+/del* mice and the frequency of GMPs is decreased (*Figure 1G,H*, *Figure 1—figure supplement 1B*), which is consistent with changes in these populations in MDS patients (*Will et al., 2012*; *Pang et al., 2013*). Thus, the *5A3* deletion perturbs HSC and myeloid progenitor populations. By contrast, the proportions and frequencies of common lymphoid progenitors are similar in WT and *5A3+/del* mice (*Figure 1—figure supplement 1C–E*).

We mixed donor *5A3+/del* or WT BM cells with WT BM at ratios of 1:1 and 1:2 and transplanted them into irradiated recipients. Whereas *5A3* mutant BM had markedly reduced lymphoid repopulating capacity, these cells efficiently contributed to the c-kit+lin−Sca-1+ (KLS) compartment (*Figure 2A,B*). To investigate if the altered repopulating potential of *5A3+/del* BM is intrinsic to CD150hi HSC, we injected 15 of these cells into lethally irradiated recipients with WT BM competitors. Similar to whole BM, CD150hi HSC from *5A3+/del* mice exhibited reduced overall repopulation potential due to defective lymphoid reconstitution (*Figure 2C–F*). Importantly, however, *5A3+/del* HSC efficiently reconstituted KLS and myeloid compartments in both primary and secondary recipients (*Figure 2E,F*). *5A3+/del* CD150hi HSC exhibited markedly reduced lymphoid repopulating potential in WT and *5A3+/del* recipients, whereas WT cells restored lymphoid repopulation in *5A3+/del* hosts almost as well as in WT recipients, demonstrating that the repopulation defects in *5A3+/del* CD150hi HSCs are cell intrinsic (*Figure 2C–F*).

WT and *5A3+/del* HSCs generated similar numbers of myeloid colonies in methylcellulose cultures supplemented with cytokines; however, colonies grown from *5A3+/del* CD150hi HSC contained significantly fewer cells (*Figure 2G,H*). By contrast, *5A3+/del* CD150neg MPP showed a 1.4-fold increase in colony forming activity but a similar number of cells per colony as WT MPP (*Figure 2G,I*). In vivo labeling experiments revealed similar rates of BrdU incorporation, cell division, and apoptosis in WT and *5A3+/del* K+L−S+ and myeloid progenitor (MP) cells (*Figure 2—figure supplement 1A,B* and data not shown) (*Nygren and Bryder, 2008*).

## Effects of aging on *5A3+/del* HSC and interaction with *Gata2* haploinsufficiency

Physiologic aging is characterized by impaired HSC repopulating potential, diminished lymphoid differentiation, the dominance of CD150hi HSCs that are skewed toward myeloid differentiation and a markedly increased risk of MDS (*Beerman et al., 2010a*; *Beerman et al., 2010b*).

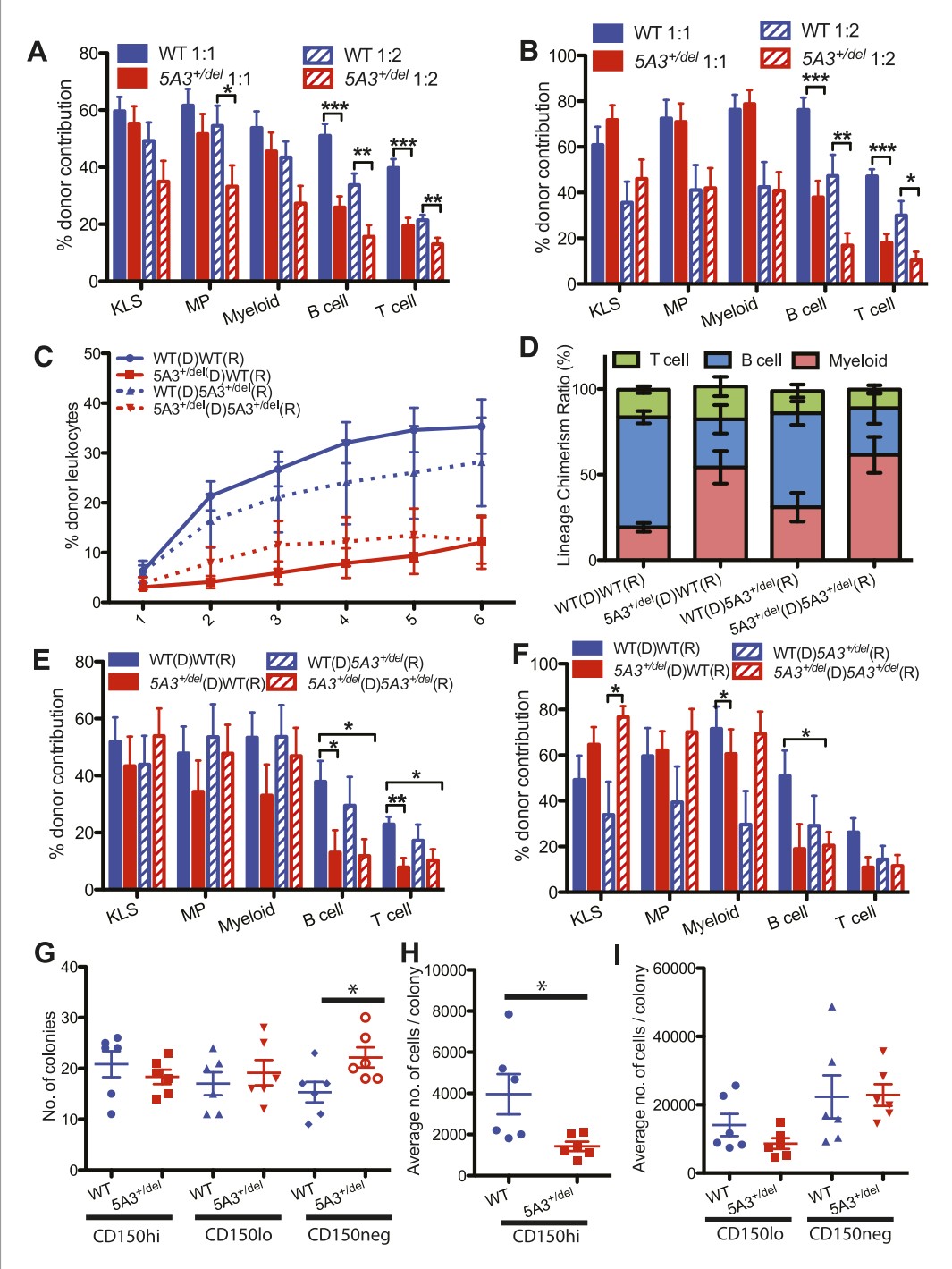

**Figure 2**. Defective repopulating potential of *5A3*[+/del] BM and CD150[hi] HSC. (**A**, **B**) BM cells from WT or *5A3*[+/del] mice (n = 9 per genotype) were mixed at ratios of 1:1 and 1:2 with WT competitor cells and transplanted into 2–3 irradiated WT recipients. Percent contribution to the K[+]L[−]S[+] (KLS), K[+]L[−]S[−] (MP), myeloid, B and T cell lineages in the BM of recipient mice 6 months after primary (**A**) and secondary (**B**) transplants are shown. (**C**) Leukocyte chimerism after competitive transplantation of 15 *5A3*[+/del] or WT CD150[hi] HSC into WT or mutant recipients (n = 12 for WT hematopoietic stem cell (HSC) in WT recipients; n = 12 for *5A3*[+/del] HSC in WT recipients; n = 8 for WT HSC in *5A3*[+/del] recipients; n = 9 for *5A3*[+/del] HSC in *5A3*[+/del] recipients). (**D**–**F**) Relative proportions of donor-derived B, T, and myeloid cells in the blood of recipient mice 6 months after transplantation (**D**). Percent contribution to the K[+]L[−]S[+] (KLS), K[+]L[−]S[−] (MP), myeloid, B and T cell lineages in the BM of recipient mice 6 months after primary (**E**) and secondary (**F**) transplants. Data shown are mean values ±SEM of results from four independent experiments.
*Figure 2. continued on next page*

*Figure 2. Continued*

(**G**–**I**) 100 CD150^hi HSC, CD150^lo HSC, and CD150^neg MPP from 8- to 12-week-old *5A3^{+/del}* mice and their WT littermates were plated into methylcellulose medium supplemented with cytokines (n = 6 for each genotype). The total number of colonies (**G**) and the average number of cells per colony (**H**, **I**) were assessed after 7 days. Dots represent individual mice, and the horizontal lines indicate median values. Data shown are mean values ±SEM of results from three independent experiments with significant differences between WT and *5A3^{+/del}* mice designated by asterisks as follows: *$p < 0.05$, **$p < 0.01$, ***$p < 0.001$.

The following figure supplement is available for figure 2:

**Figure supplement 1**. Proliferation and divisional kinetics of *5A3^{+/del}* HSCs.

Similarly, the abnormal distribution of HSCs is exacerbated in 24- to 30-month-old *5A3^{+/del}* mice (*Figure 3A,B*). Consistent with data from younger mice (*Figure 2A–F*), aged *5A3* mutant BM displayed markedly reduced lymphoid repopulating potential, but efficiently contributed to myeloid reconstitution (*Figure 3C,D*). Old *5A3^{+/del}* BM cells also repopulated the KLS compartment significantly better than WT BM in recipients analyzed 4 months after transplantation, and *5A3^{+/del}* cells exhibited a twofold increase in contribution to the KLS and MP populations upon serial transplantation (*Figure 3C,D*). Despite these HSC abnormalities, *5A3^{+/del} mice* have a normal lifespan, and the underlying causes of death are similar to WT littermates (data not shown).

GATA2 mutations cause familial MDS, which is frequently characterized by −7/del(7q) (*Hahn et al., 2011*; *Bodor et al., 2012*; *Pasquet et al., 2013*). We generated *Gata2^{+/−}*; *5A3^{+/del}* mice, quantified HSC and progenitor populations, and performed competitive repopulation experiments. Like *5A3^{+/del}* mutant mice, *Gata2^{+/−}*; *5A3^{+/del}* mice have reduced BM cellularity and spleen weights (*Figure 3E,F*), and the CD150^hi HSC bias is augmented by concomitant *Gata2* deletion (*Figure 3G*). Whereas lymphoid reconstitution is similarly impaired in *5A3^{+/del}* and compound mutant mice, the *5A3* deletion partially rescues the repopulating deficit of *Gata2^{+/−}* BM in the KLS, MP, and myeloid compartments (*Figure 3H*) (*Rodrigues et al., 2005*). These data suggest that the 7q22 CDS contributes to transformation in familial MDS by impairing lymphoid differentiation while also modestly enhancing the growth of *GATA2* mutant HSC and their myeloid progeny.

## Changes in gene expression and metabolic activities in 5A3^{+/del} hematopoietic cells

Transcriptome (RNA-Seq) and TaqMan quantitative real-time PCR analyses revealed a ~50% reduction in the expression of multiple genes and of the long intergenic non-coding RNA 503142E22Rik within the *5A3* interval in mutant HSC and MPP (*Figure 4A,B*). Gene Set Enrichment Analysis (GSEA) of the RNA-Seq data from *5A3^{+/del}* HSCs further demonstrated reduced expression of multiple gene sets related to oxidative phosphorylation (OXPHOS) that are similarly down-regulated in the early stages of human therapy-induced MDS and AML (*Figure 4C*) (*Mootha et al., 2003*; *Li et al., 2011*). OXPHOS is the metabolic pathway used by cells to generate adenosine triphosphate (ATP). OXPHOS is regulated by mitochondrial membrane potential, and defects in this metabolic pathway can increase levels of reactive oxygen species (ROS). However, sorted WT and *5A3^{+/del}* HSC and MPP showed similar ATP levels (*Figure 4D*). Membrane potential and intracellular ROS levels were also similar in HSC and MPP from young mice, but ROS levels were increased by ~50% in the HSC and MPP of aged *5A3^{+/del}* mice (*Figure 4E,F*). Elevated ROS levels in HSCs correlate with reduced self-renewal capacity, impaired multi-lineage repopulating ability, and myeloid-biased differentiation (*Jang and Sharkis, 2007*). ROS levels are also increased in t-MDS/AML BM (*Reinecke et al., 2009*; *Li et al., 2011*).

Down-regulation of OXPHOS genes is expected to reduce mitochondrial respiration in HSC (*Warr and Passegue, 2013*). We attempted to directly measure oxygen consumption rates (OCRs) in CD150^hi and CD150^lo HSC but could not obtain reproducible results due to limiting cell numbers. We therefore compared the OCRs of KLS, MP, B, and T cells isolated from 1-year-old WT and *5A3^{+/del}* mice. *5A3^{+/del}* KLS cells showed a similar basal OCR as their WT counterparts, but a slightly lower maximal respiratory capacity that did not reach statistical significance (p = 0.2562) (*Figure 4G*). *5A3^{+/del}* MPs surprisingly showed a significantly higher basal respiration and maximal

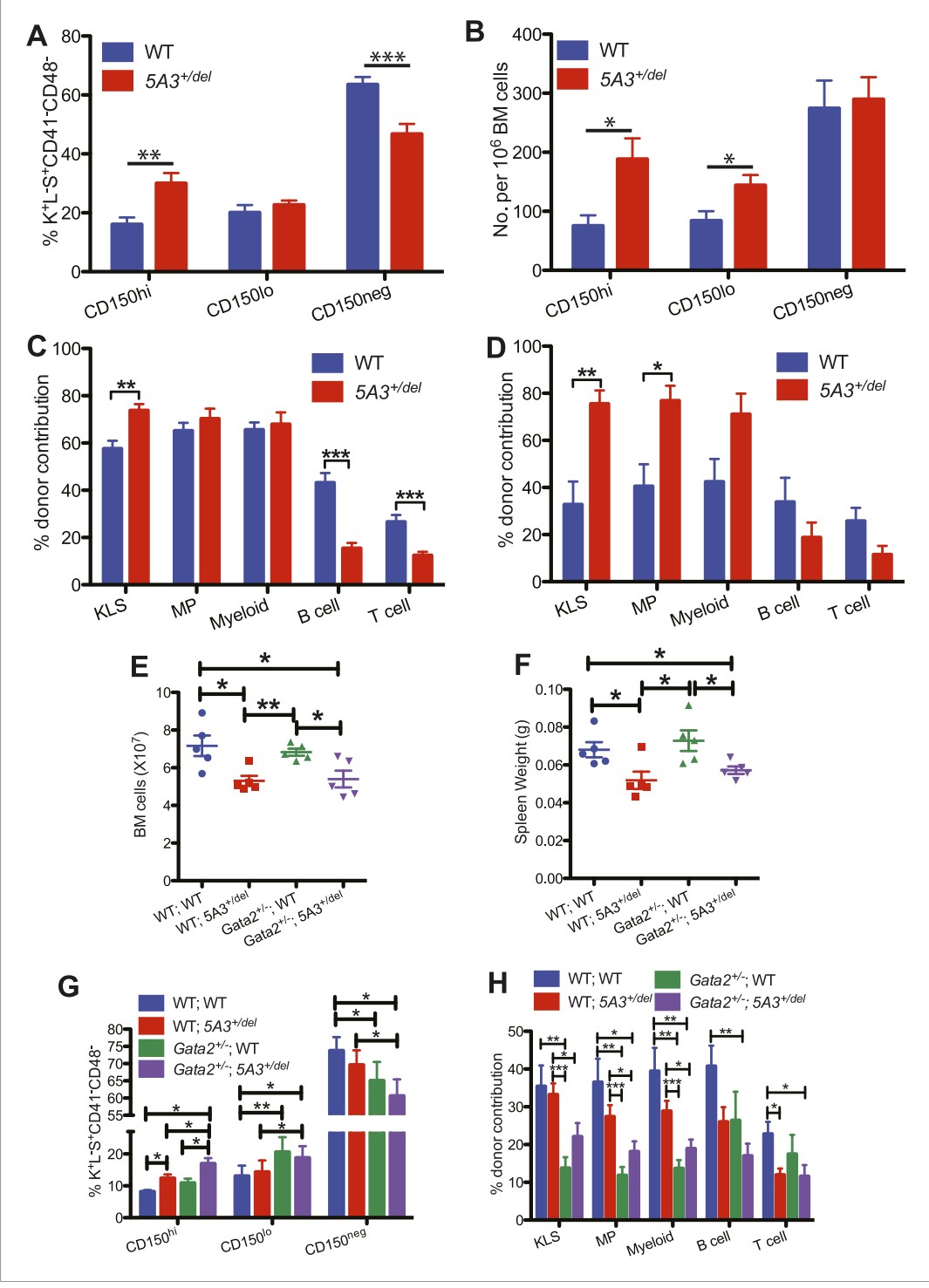

**Figure 3**. Effects of aging and *Gata2* inactivation on *5A3*[+/del] hematopoietic cells. (**A** and **B**) Percent contributions (**A**) and frequencies (**B**) of CD150[hi] HSC, CD150[lo] HSC, and CD150[neg] MPP within the K[+]L[−]S[+]CD41[−]CD48[−] compartment in *5A3*[+/del] mice (n = 12) and their WT littermates (n = 11) at 24–30 months of age. (**C** and **D**) Competitive transplantation of BM cells from 24- to 30-month-old WT or *5A3*[+/del] mice. Percent donor contribution of *5A3*[+/del] cells to K[+]L[−]S[+] (KLS), K[+]L[−]S[−] (MP), myeloid, B cell, and T cell populations in the BM of recipient mice 4 months after primary (**C**) and secondary (**D**) competitive transplantation (n = 5 donors and 10 recipients of each genotype). Data shown are mean values ±SEM of results from two independent experiments. (**E**) Total numbers of BM cells from 2 femurs and 2 tibiae in *5A3*[+/del] mice, *Gata2*[+/−] mice, compound *Gata2*[+/−]; *5A3*[+/del] mice and WT

*Figure 3. continued on next page*

*Figure 3. Continued*

littermates at 8–12 weeks of age. (**F**) Spleen weights in *5A3*[+/del] mice, *Gata2*[+/−] mice, compound *Gata2*[+/−]; *5A3*[+/del] mice, and WT littermates at 8–12 weeks of age. (**G**) Percent contributions of cells with high (CD150[hi] HSC), low (CD150[lo] HSC), and absent CD150 expression (CD150[neg] MPP) within the K[+]L[−]S[+]CD41[−]CD48[−] compartment of WT, *5A3*[+/del], *Gata2*[+/−], and compound *Gata2*[+/−]; *5A3*[+/del] mice at 8–12 weeks of age (n = 5 of each genotype). (**H**) BM cells from WT, *5A3*[+/del], *Gata2*[+/−] or compound *Gata2*[+/−]; *5A3*[+/del] mice (n = 5 of each genotype) were each mixed at ratios of 1:1 with WT competitor cells and transplanted into two irradiated WT recipients. Percent contribution to the K[+]L[−]S[+] (KLS), K[+]L[−]S[−] (MP), myeloid, B and T cell lineages in the BM of recipient mice 6 months after primary transplants. Data shown are mean values ±SEM from five independent experiments with significant differences designed by asterisks as follows: *p < 0.05, **p < 0.01, ***p < 0.001. The enhanced repopulating ability of compound *Gata2*[+/−]; *5A3*[+/del] vs *Gata2* singly mutant HSC achieved borderline statistical significance in three myeloid populations (KLS (p = 0.09), MP (p = 0.09), and myeloid cells (p = 0.12)).

respiratory capacity in comparison to WT cells, supporting an overall increase in energy consumption (*Figure 4H*). Meanwhile, mature *5A3*[+/del] B and T cells have similar mitochondrial stress profiles as WT cells. We conclude that global changes in OXPHOS gene expression exert a modest impact on cellular metabolism in aged *5A3* mutant HSC and progenitors. Interestingly, treatment with N-acetyl-L-cysteine (NAC) did not reverse the hematopoietic abnormalities in young *5A3*[+/del] mice (data not shown), suggesting that they are a direct consequence of the *5A3* deletion and are not secondary to ROS production.

Segmental deletions are among the most frequent genetic alterations in human cancer, and simultaneous loss of multiple haploinsufficient TSGs that individually have minimal phenotypic consequences appears to underlie the growth advantage conferred by most of these chromosomal losses (*Solimini et al., 2012*). We show that a haploinsufficient deletion in mice that models loss of a human 7q22 CDS causes hematopoietic abnormalities that include a myeloid lineage output bias, impaired lymphoid repopulating potential, and a pronounced age-associated expansion of HSC and MPs. These functional abnormalities support a role of 7q22 deletions in MDS pathogenesis. The seven genes within the deleted *5A3* segment that are expressed in HSC encode proteins that regulate diverse cellular processes including transcription (*Mll5* and *Dnajc2*), mitochondrial quality control (*Pmpcb, Armc10*), protein degradation (*Psmc2*), biosynthesis of *N*-acylethanolamines (*Napepld*), and DNA replication (*Orc5*) (*Luciano et al., 1997*; *Quintana et al., 1998*; *Fujiki et al., 2009*; *Sebastian et al., 2009*; *Richly et al., 2010*; *Nijhawan et al., 2012*; *Tsuboi et al., 2013*; *Zhou et al., 2013*; *Serrat et al., 2014*). Given this, it is likely that the haploinsufficiency for multiple interacting genes leads to altered hematopoietic differentiation and function in *5A3*[+/del] mice. Similar to the *5A3*[+/del] mice described here, other mutations found in MDS patients perturb hematopoiesis, but do not consistently induce hematologic disease in the absence of cooperating mutations (*Beurlet et al., 2013*). This is not unexpected given the advanced age of most MDS patients and the presence of multiple genetic lesions in diseased BM. The *5A3*[+/del] mice reported here thus provide a novel resource for addressing how this common deletion cooperates with other mutations to drive myeloid transformation, progression to AML, and drug resistance.

## Materials and methods

### Mice

We expressed Cre recombinase in embryonic stem cells containing *LoxP* sites and hypoxanthine phosphoribosyl transferase (HPRT) sequences flanking the *Fbxl13* and *Srpk2* genes (*Wong et al., 2010*) and analyzed clones that grew in hypoxanthine-aminopterin-thymidine (HAT) medium to identify the desired 2 Mb deletion. *5A3*[del/+] mice were generated by standard blastocyst injection followed by mating coat color chimeras and were backcrossed for at least 10 generations onto a C57BL/6J background. *Gata2*[+/−] mice were a generous gift from Dr Stuart Orkin (Harvard Medical School) (*Tsai et al., 1994*). Study mice were housed in a specific pathogen-free facility at the University of California San Francisco, and all animal experiments were conducted under protocols approved by the Institutional Animal Care and Use Committee. Genotyping and disease monitoring were performed as previously described (*Wong et al., 2010*).

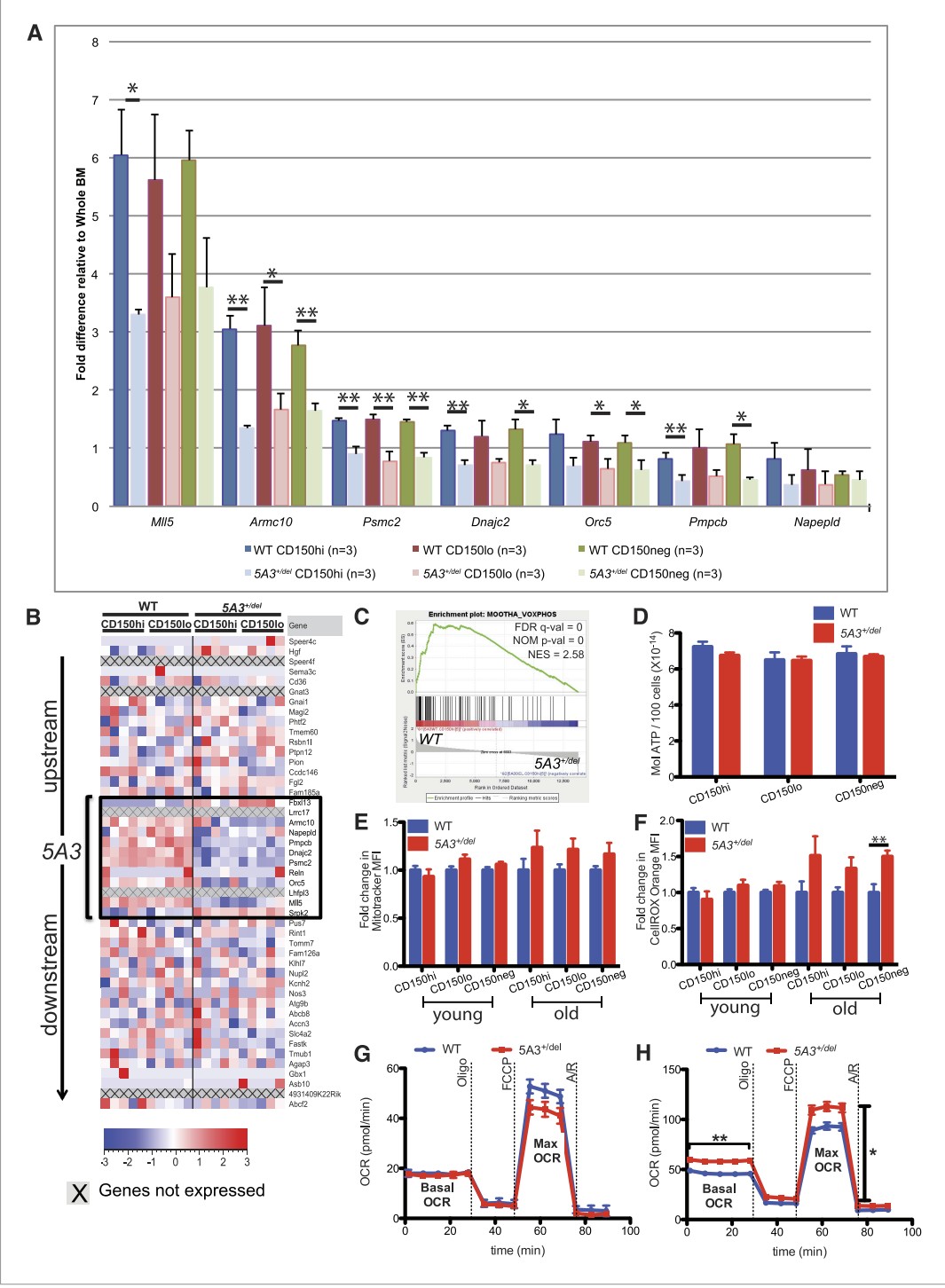

**Figure 4**. Changes in gene expression and oxidative phosphorylation in *5A3*[+/del] HSC and MP. (**A**) Relative mRNA abundances for genes within the deleted *5A3* interval expressed at detectable levels in sorted HSC populations were determined by TaqMan reverse transcriptase PCR (n = 3 per genotype). (**B**) Expression levels of genes located within and flanking the deleted interval measured by RNA-Seq in sorted CD150hi HSC and CD150lo HSC from 5 mice of each genotype. Each column presents data from an individual mouse, and genes within the *5A3* deleted region are delimited with a black box. Three non-coding RNAs (6030443J06Rik, AC112688.1 and 5031425E22Rik) are located within the *5A3* deletion. Two of these (6030443J06Rik and AC112688.1) are expressed at extremely low levels in HSC, and the other (5031425E22Rik) showed ~50% lower expression in *5A3*[+/del] HSC. 5031425E22Rik is homologous to the human *KMT2E* (a.k.a. *MLL5*) antisense RNA1. Expression levels of the flanking *Fbxl13* and *Srpk2*

*Figure 4. Continued*

genes are modestly up-regulated in *5A3*[+/del] HSC, which is consistent with the targeting strategy used to generate the segmental deletion. (**C**) Gene Set Enrichment Analysis of *5A3*[+/del] CD150[hi] HSCs revealed negative enrichment for genes associated with oxidative phosphorylation (OXPHOS). False discovery rate (FDR) q-val, nominal p-value (NOM p-value), and normalized enrichment scores (NESs) are indicated. (**D**) ATP levels in HSC and MPP from 8- to 12-week-old WT (n = 6) and *5A3*[+/del] (n = 5) mice. Data shown are mean values ±SEM of results from two independent experiments. (**E**) Fold change in the mean MitoTracker Orange fluorescence levels in *5A3*[+/del] cells normalized to values in WT cells analyzed in the same experiment. (**F**) Fold change in the mean fluorescence level (MFI) of *5A3*[+/del] cells that are CellROX Orange positive normalized to values in WT cells analyzed in the same experiment. For the MitoTracker and CellROX experiments, n = 13 for WT and n = 12 for *5A3*[+/del] young mice, three independent experiments; n = 5 for WT and n = 6 for *5A3*[+/del] aged mice, two independent experiments. Data shown are mean values ±SEM of results from independent experiments. (**G** and **H**) Oxygen consumption rate (OCR) was assessed basally and in response to the mitochondrial inhibitors oligomycin (oligo), carbonyl cyanide 4-(trifluoromethoxy) phenylhydrazone (FCCP), and antimycin A and rotenone (A/R) for (**G**) KLS and (**H**) MP cells. Data are shown as mean ±SEM of n = 5 mice of each genotype from two independent experiments.

## Flow cytometry

BM cells flushed from tibias and femurs were subjected to ammonium-chloride potassium red cell lysis before staining with antibodies. For experiments requiring cell sorting, the spinal cord, flat bone of the pelvis, humerus, and sternum of the mice were also crushed and lysed. Low-density mononuclear cells were separated using a HISTOPAQUE-1119 gradient (Sigma–Aldrich, St. Louis, MO, United States). For identification and sorting of CD150[hi]-HSC, CD150[lo]-HSC, and CD150[neg]-MPP, cells were pre-incubated with purified CD16/32 (2.4G2), followed by staining with a lineage cocktail of PE-conjugated antibodies including B220 (RA3-6B2), CD8 (53-6.7), Gr-1 (RB6-8C5), CD3 (17A2), Ter119 (TER-119), CD41 (MWReg30), and CD48 (HM48-1), as well as PE-Cy7 c-kit (2B8), PacBlue Sca-1 (E13-161.7), and APC CD150 (TC15-12F12.2) from BioLegend (San Diego, CA, United States). For experiments requiring cell sorting, cells expressing c-kit were enriched by magnetic cell sorting by staining with mouse CD117 microbeads and positively selected on the MS columns (Miltenyi Biotec, Germany) according to manufacturer's protocol before antibody staining. Cells were classified as CD150[hi]-HSC, CD150[lo]-HSC, or CD150[neg]-MPP based in levels of CD150 expression.

Percent contribution of common myeloid progenitor (CMP), granulocyte-monocyte progenitor (GMP), and megakaryocyte erythroid progenitors (MEP) within the Lin−Sca1+c-kit+ compartment of 5A3+/del mice and WT littermates. CMP, GMP, and megakaryocyte erythroid progenitors (MEP) populations were identified by flow cytometry after staining with a lineage cocktail of PE-conjugated antibodies including B220 (RA3-6B2), CD8 (53-6.7), Gr-1 (RB6-8C5), CD3 (17A2), and Ter119 (TER-119), as well as APC-750 c-kit (2B8), PacBlue Sca-1 (E13-161.7), Alexa Fluor 647 CD16/32 (93), biotin CD34 (MEC14.7), and a streptavidin PE-Cy7 conjugate (BioLegend). Stained BM cells were analyzed in a FACS LSRII instrument and sorted in a FACS Aria3 (Becton Dickinson, San Jose, CA, United States). FlowJo software (Tree Star, Inc., Ashland, OR, United States) was used to analyze and display the data.

## Competitive repopulation

BM cells from WT, *5A3*[+/del], *Gata2*[+/−], *Gata2*[+/−]*5A3*[+/del] mice on a C57BL/6J background (CD45.2) were used as donor cells. Competitor cells were isolated from 8- to 12-week-old BoyJ mice (CD45.1). Recipient F1 hybrid mice from a cross between C57BL/6J and BoyJ mice (CD45.1 + CD45.2) were at least 8 weeks old at the time of lethal irradiation (9.5 Gy from a cesium source delivered in split dose 3 hr apart). After irradiation, the cells were injected via the tail vein of recipient mice. For evaluation of the competitiveness of whole BM, we injected $10^6$ whole BM cells at a 1:1 or 1:2 donor to competitor ratio. To evaluate the repopulating potential of purified CD150[hi]-HSC, we injected 15 CD150[hi]-HSC sorted from 8- to 12-week-old WT and *5A3*[+/del] mice together with $2.5 \times 10^5$ BM competitor cells into lethally irradiated recipients.

Blood was obtained from recipient mice every 30 days beginning 1 month after transplant, and cells were stained with Alexa Fluor 700 CD45.2 (104), PE-Cy7 CD45.1 (A20), PacBlue B220 (RA3-6B2),

FITC CD4 (GK1.5), FITC CD8 (53-6.7), PE Mac-1 (M1/70), and PE Gr-1 (RB6-8C5) to determine the percent donor cell contribution to myeloid and B and T lymphoid lineages. Primary recipient mice were euthanized 6 months after transplantation, BM were isolated from the tibiae and femur, and $2 \times 10^6$ BM cells were injected into secondary recipients to test serial repopulation potential. To determine the contribution of donor-derived cells in the $K^+L^-S^+$ compartment, BM cells were stained with V450 CD45.2 (104), APC-780 CD45.1 (A20), PE B220 (RA3-6B2), PE CD8 (53-6.7), PE Gr-1 (RB6-8C5), PE CD3 (17A2), PE Ter119 (TER-119), PE-Cy7 c-kit (2B8), and APC Sca-1 (E13-161.7). Secondary recipients were analyzed as in the primary recipients.

## Methylcellulose colony assays

CD150hi-HSC, CD150lo-HSC, and CD150neg-MPP were isolated as described above, and 100 cells were seeded into methylcellulose medium as described (*Mohrin et al., 2010*). Colonies were counted on day 7, and the entire contents of a methylcellulose culture from an individual plate were then flushed out using phosphate-buffered saline and counted in a hemocytometer. Cells were also spun in a Cytospin 3 Cytocentrifuge (Shandon/Thermo Fisher Scientific, Waltham, MA, United States) at 400 rpm for 8 min, and differential cell counting and morphological analysis performed after Wright-Giemsa staining.

## RNA isolation and expression

CD150hi-HSC, CD150lo-HSC, and CD150neg-MPP were sorted as above into 500 µl of TRIzol reagent (Life Technologies, Grand Island, NY, United States). RNA was isolated according to manufacturer's instructions and precipitated with the addition of glycogen (New England BioLabs, Ipswich, MA, United States). The RNA was then treated with DNAse 1 (Ambion, Austin, TX, United States) and purified with the RNeasy MinElute Cleanup Kit (Qiagen, Valencia, CA, United States).

For TaqMan analysis, reverse transcription was carried out using the High Capacity RNA-to-cDNA Master Mix (Life Technologies). Relative quantification of gene expression was determined by performing quantitative real-time PCR using the following TaqMan Gene Expression Assays (Applied Biosystems): *Armc10* (Mm03011576_g1), *Mll5* (Mm01129502_g1), *Napepld* (Mm00724596_m1), *Orc5* (Mm00457242_m1), *Psmc2* (Mm00803207_m1), *Dnajc2* (*Mm00494389_m1*), *Pmpcb* (*Mm01138654_m1*), *Srpk2* (*Mm00486413_m1*), *Fbxl13* (*Mm00622025_m1*), *Lrrc17* (*Mm01167263_m1*), *Slc26a5* (*Mm00446145_m1*), *Reln* (Mm00465200_m1), *Lhfpl3* (Mm03038441_m1), and *Gapdh* (Mm99999915_g1) with the TaqMan Gene Expression Master Mix (ABI). PCR reactions were performed on an ABI 7900HT Real-Time PCR System (Applied Biosystems, Foster City, CA, United States) with the Taqman Gene Expression Master Mix (Applied Biosystems) according to manufacturer's instructions. PCR cycling conditions were 2 min at 50°C and 10 min at 95°C, followed by 40 cycles of 15 s at 95°C and 1 min at 60°C. All reactions were carried out in triplicate, and target quantities were determined using a relative standard curve. The amounts of target were normalized to the endogenous control gene *Gapdh* and compared with the corresponding WT BM (calibrator sample) to determine relative fold differences.

For RNA-Seq analysis, total RNA (10 ng) was converted into double-stranded cDNA using the Ovation RNA Amplification System V2 (NuGen, San Carlo, CA, United States) per manufacturer's recommendations. The amplified cDNA products were then used to generate RNA-seq libraries using the TruSeq RNA Sample Preparation Kit v2 reagents (Illumina, San Diego, CA, United States) per manufacturer's instructions, with 10 PCR amplification cycles. Library quality and quantity were assessed by the Agilent DNA1000 Chip (Agilent, Santa Clara, CA, United States) and qPCR (Kappa Biosystems Inc, Woburn, MA, United States). 10 pM of each library was sequenced using Illumina SBS chemistry at $2 \times 100$ bp reads on the HiSeq2000 (Illumina, San Diego, CA, United States).

The RNA-Seq paired-end reads were mapped to the mouse mm9 genome using an in-house mapping and quality assessment pipeline (*Zhang et al., 2012*). The expression of each gene was estimated by the mean coverage of the highest covered coding exon. Genes with low-expression level (<10) across all samples were filtered out, followed by quantile normalization. Differential expression analysis was performed using limma (*Smyth, 2004*) with estimation of false discovery rate (*Benjamini and Hochberg, 1995*). GSEA (*Subramanian et al., 2005*, *2007*) was used to assess pathway enrichment.

## ATP quantification

HSC and MPP were sorted into phosphate buffered saline (PBS), and 600 CD150$^{hi}$-HSC, 1000 CD150$^{lo}$-HSC, and 1000 CD150$^{neg}$-MPP were aliquoted into a well of a 96-well plate in triplicate. ATP was quantified using the CellTiter-Glo Luminescent Cell Viability Assay (Promega, Madison, WI, United States) following manufacturer's recommendations. Illumination was quantified with a Synergy 2 (Biotek, Winooski, VT, United States).

## Metabolic studies

For flow cytometric analysis of ROS levels and membrane potential in CD150$^{hi}$-HSC, CD150$^{lo}$-HSC, and CD150$^{neg}$-MPP, BM cells were isolated and enriched for c-kit positive cells as described. c-kit positive cells were stained with an unconjugated cocktail of purified antibodies including B220 (RA3-6B2), CD8 (53-6.7), Gr-1 (RB6-8C5), CD3 (17A2), Ter119 (TER-119), and CD41 (MWReg30) from BioLegend, followed by PECy-5 IgG GOAT anti-RAT pAb (HI47) antibody from Molecular Probes and IgG from rat serum (I4131 Sigma). After incubation with 50 nM CellROX Orange Reagent (Molecular Probes) or 20 nM MitoTracker Orange CMTMRos (Molecular Probes, Eugene, OR, United States) for 30 min at 37°C in IMDM, cells were washed and stained with 7-amino-actinomycin D, PE-Cy7 c-kit (2B8), PacBlue Sca-1 (E13-161.7), PE-Cy5 CD48 (HM48-1), and APC CD150 (TC15-12F12.2) from BioLegend.

The OCR was analyzed in an XF96 extracellular flux analyzer following manufacturer's protocol (Seahorse Biosciences, Billerica, MA, United States). Freshly isolated K$^+$L$^-$S$^+$ cells and K$^+$L$^-$S$^-$ cells were cultured in StemSpan serum-free medium (STEMCELL Technologies, Vancouver, Canada) supplemented with SCF (100 ng/ml) and Tpo (100 ng/ml), while freshly isolated thymic cells and B-220$^+$ splenic cells were cultured in RPMI medium 1640 supplemented with 10% fetal calf serum and incubated at 37° in a humidified atmosphere containing 8% $CO_2$ for 12–15 hr. Cells were then washed three times with Mito Stress Media (XF base media supplemented with glucose [3 mg/ml], sodium pyruvate [1 mM], and Glutamax [2 mM] adjusted to a pH = 7.4) and seeded in XF96 microplates coated with poly-L-lysine (Sigma). 60000 K$^+$L$^-$S$^+$ cells, 100000 K$^+$L$^-$S$^-$ cells, 200000 thymic cells, and 200000 B220$^+$ splenic cells were plated per well, respectively. K$^+$L$^-$S$^+$ and K$^+$L$^-$S$^-$ cells were stimulated with SCF (100 ng/ml) and Tpo (100 ng/ml), while thymic cells were stimulated with interleukin (IL)-2 (20 ng/ml) and IL-7 (10 ng/ml) and B-220$^+$ splenic cells were stimulated with IL-7 (10 ng/ml) and maintained in a non-$CO_2$ incubator for 1 hr before the assay. Five baseline recordings were made, followed by sequential injection of Oligomycin (Sigma), Carbonyl Cyanide 4-(trifluoromethoxy)phenylhydrazone (Sigma), and a combination of Antimycin A (Sigma) and Rotenone (Sigma) to determine the mitochondrial respiration rate under various conditions.

## Statistical analysis

Data are presented as mean values ±SEM unless stated otherwise. Statistical significance was determined by performing two-tailed Student's *t*-tests.

## Antibodies for flow cytometry of CLP cells

CLP were identified and evaluated by flow cytometry by staining with a lineage cocktail of PE-conjugated antibodies including B220 (RA3-6B2), CD8 (53-6.7), Gr-1 (RB6-8C5), CD3 (17A2), and Ter119 (TER-119), as well as APC-750 c-kit (2B8), PacBlue Sca-1 (E13-161.7), APC IL-7Rα (A7R34) and biotin Flk2 (A2F10), and a streptavidin PE-Cy7 conjugate (BioLegend).

## BrdU staining

Mice received an initial intraperitoneal injection of BrdU (Sigma–Aldrich, 1 mg/6 g mouse weight) and were then maintained on 1.0 mg/ml BrdU in the drinking water 24 hr prior to sacrifice. To measure BrdU incorporation, BM cells were enriched for c-kit positive cells using c-kit antibody-conjugated microbeads (Miltenyi). Enriched BM cells were pre-incubated with purified CD16/32 (2.4G2), followed by staining with a lineage cocktail of PE-conjugated antibodies including B220 (RA3-6B2), CD8 (53-6.7), Gr-1 (RB6-8C5), CD3 (17A2), Ter119 (TER-119), as well as PECy7 c-kit (2B8) and PacBlue Sca-1 (E13-161.7) (BioLegend). BrdU incorporation was then assayed according to manufacturer's instructions of the APC BrdU Flow Kit (BD Pharmingen, San Jose, CA, United States).

### NHS-biotin staining

NHS-biotin (EZ-Link Sulfo-NHS-LC-LC-biotin; Life Technologies) was dissolved at 10 mg/ml in normal saline and injected into mice intravenously via the tail vein at 1 mg/6 g mouse weight. To assess biotin dilution in the HSC subpopulations, BM cells were enriched for c-kit positive cells using c-kit antibody-conjugated microbeads (Miltenyi). Enriched cells were pre-incubated with purified CD16/32 (2.4G2), followed by staining with a lineage cocktail of PE-conjugated antibodies including B220 (RA3-6B2), CD8 (53-6.7), Gr-1 (RB6-8C5), CD3 (17A2), Ter119 (TER-119), CD41 (MWReg30), and CD48 (HM48-1), as well as PerCP/Cy5.5 c-kit (2B8), PacBlue Sca-1 (E13-161.7), APC CD150 (TC15-12F12.2), and PECy-7-conjugated Streptavidin (BioLegend).

## Acknowledgements

We are grateful to Michelle Le Beau, Qing Li, Scott Lowe, and Megan McNerney for insightful suggestions and to Stuart Orkin for providing the $Gata2^{+/-}$ mutant mice. This work was supported by NIH grants P01 CA40046, R37 CA72614, T32 CA108462, and R01 HL092471, by the Kenneth and Mary Ellen Wilson St. Baldrick's Research Grant, by a Specialized Center of Research award from the Leukemia and Lymphoma Society (LLS 7019-04), and by the ALSAC of St. Jude Children's Research Hospital. KS is an American Cancer Society Research Professor, MRB is supported by the American Cancer Society Hillcrest Fellowship, and EP is a Leukemia and Lymphoma Society Scholar.

## Additional information

### Funding

| Funder | Grant reference | Author |
|---|---|---|
| National Cancer Institute (NCI) | T32 CA108462, PO1 CA40046, R37 CA72614 | Michael R Burgess, James R Downing, Kevin Shannon |
| St. Baldrick's Foundation | Kenneth and Mary Ellen Wilson St. Baldrick's Research Grant 123584A | Kevin Shannon |
| Leukemia and Lymphoma Society (LLS) | Specialized Center of Research Award LLS 7019-04 | James R Downing, Kevin Shannon |
| St. Jude Children's Research Hospital | American Lebanese Syrian Associated Charities | James R Downing |
| National Heart, Lung, and Blood Institute (NHBLI) | RO1 HL092471 | Emmanuelle Passegué |
| American Cancer Society | Research Professorship | Kevin Shannon |
| American Cancer Society | Hillcrest Fellowship | Michael R Burgess |

The funders had no role in study design, data collection and interpretation, or the decision to submit the work for publication.

### Author contributions

JCW, KMW, Conception and design, Acquisition of data, Analysis and interpretation of data, Drafting or revising the article; MA, JN, JC, JF, DR, Acquisition of data, Analysis and interpretation of data, Drafting or revising the article; SCK, JM, S-CC, Analysis and interpretation of data, Drafting or revising the article; MRB, Acquisition of data, Drafting or revising the article, Contributed unpublished essential data or reagents; YZ, Drafting or revising the article, Contributed unpublished essential data or reagents; TTH, Acquisition of data, Drafting or revising the article; EP, JRD, KS, Conception and design, Analysis and interpretation of data, Drafting or revising the article

### Ethics

Animal experimentation: Study mice were housed in a specific pathogen-free facility at the University of California San Francisco, and all animal experiments were conducted in strict accordance with the protocols approved by the Institutional Animal Care and Use Committee (IACUC) of the University of California, San Francisco (Approval number: AN091877-03).

## Additional files

### Major dataset

The following dataset was generated:

| Author(s) | Year | Dataset title | Dataset ID and/or URL | Database, license, and accessibility information |
|---|---|---|---|---|
| Wong JC, Weinfurtner KM, Alzamora M, Kogan SC, Burgess MR, Zhang Y, Nakitandwe J, Ma J, Cheng J, Chen S, Ho TT, Flach J, Reynaud D, Passegue E, Downing JR, Shannon K | 2015 | Data from: Functional evidence implicating chromosome 7q22 haploinsufficiency in myelodysplastic syndrome pathogenesis | http://www.ncbi.nlm.nih.gov/geo/query/acc.cgi?acc=GSE72811 | Available at Gene Expression Omnibus (GEO) with the accession number GSE72811. |

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
