## [Decision Letter]

Thank you for submitting your work entitled “Functional evidence implicating chromosome 7q22 haploinsufficiency in myelodysplastic syndrome pathogenesis” for peer review at *eLife*. Your submission has been favorably evaluated by Fiona Watt (Senior editor) and three reviewers, one of whom, Chi Dang, is a member of our Board of Reviewing Editors.

The reviewers have discussed the reviews with one another and the Reviewing editor has drafted this decision to help you prepare a revised submission.

Summary:

The manuscript by Wong et al. reports an intriguing mouse model of human 7q deletion, *5A3(+/del)* (termed *5A3del*, herein), which is associated with myelodysplastic syndrome (MDS). The mice were smaller in size than their wild-type (WT) counterparts. They documented defects of the *5A3del* hematopoietic stem cell (HSC) KLS-CD150hi cells, which were relatively increased in mutant mice as compared with WT mice. These findings indicate that there is a relative expansion of these HSC pools along with the finding that common myeloid progenitors were increased. The granulocyte-monocyte progenitors were decreased. The *5A3del* HSCs were also diminished in lymphoid progenitors. The authors further characterize mutant versus WT HSCs in reconstitution and transplant models, which all support defects seen in human MDS. However, similar to other mouse models of MDS-associated gene mutations, their mice do not display the hallmarks of MDS; namely, cytopenia, dysplasia, or transformation to AML were not observed. Convincing evidence is provided that there is a cell-autonomous shift in lineage bias of HSCs towards myeloid differentiation. Models were used to study aging HSCs and the authors uncovered a myeloid skewing of *5A3del* HSCs versus WT HSCs. Further, gene expression analysis indicates that genes involved in oxidative phosphorylation (OXPHOS) were diminished. Direct measurements also revealed elevated reactive oxygen species (ROS) levels in aging mutant HSCs. Intriguingly, N-acetylcysteine did not rescue the hematopoietic defects of mutant HSCs. Overall, the model generated appears to recapitulate human MDS associated with chromosome 7q deletion. The finding on alterations of OXPHOS could open up a new avenue of research to further our understanding of the pathogenesis of MDS. The manuscript in its current form, however, did not relate the phenotypic changes with genes that are involved by chromosome 7 deletion. Hence, the mechanistic insight provided by the model lacks depth. Nonetheless, this is an important study of the hematologic consequences of heterozygous inactivation of the 7q22 common deleted region defined for del(7q) MDS.

Essential revisions:

For readers who are less familiar with the genetic architecture of 7q deletions in myeloid malignancies, it would be helpful to provide a broader discussion (in the Introduction or Discussion) of the critical genes on 7q now that a number have been identified that are recurrently mutated. Listing the genes that are mutated, and the frequency of 7q22 loss in the absence of some/all of these genes would be helpful to understand how the 7q22 interval cooperates with other 7q genes and the expected phenotype of the 7q22 deletion in isolation.

Specifically, for Figure 1, it is suggested that the authors include a schematic showing the commonly deleted regions on human chromosome 7. This will help readers place the current work in context.

For example, whether *EZH2* alterations have been linked to changes in OXPHOS gene expression was not addressed and could be briefly discussed. Given the finding about OXPHOS genes and ROS, ultrastructural analysis of the HSCs for changes in mitochondrial structure might be instructive but optional for this Short Report format.

The elevated CD150 hi population and the altered OxPhos expression raises the question of the state of quiescence of the HSCs. The manuscript notes unchanged BrdU incorporation (data not shown). It would be helpful to show a cell cycle analysis of the stem cell population if possible.

---

## [Author Response]

*For readers who are less familiar with the genetic architecture of 7q deletions in myeloid malignancies, it would be helpful to provide a broader discussion (in the Introduction or Discussion) of the critical genes on 7q now that a number have been identified that are recurrently mutated. Listing the genes that are mutated, and the frequency of 7q22 loss in the absence of some/all of these genes would be helpful to understand how the 7q22 interval cooperates with other 7q genes and the expected phenotype of the 7q22 deletion in isolation*.

*Specifically, for*
Figure 1*, it is suggested that the authors include a schematic showing the commonly deleted regions on human chromosome 7. This will help readers place the current work in context*.

These are excellent suggestions that speak to the central point that the DNA segment that we used chromosome engineering to excise is of interest because of its strong association with myeloid malignances, particularly myelodysplastic syndrome (MDS) and acute myeloid leukemia (AML). We modified the Introduction and Figure 1 to underscore this point and to concisely summarize the current state of knowledge regarding 7q genes within and outside the relevant 7q22/5A3 interval mutated in MDS and AML. The revised text appears in the second paragraph of the Introduction, and Figure 1 now presents a diagram of human chromosome 7q with genes implicated as candidate myeloid tumor suppressors shown in italics above, and commonly deleted segments indicated with brackets immediately below. The legend to Figure 1 has been revised. Importantly, targeted and genome-wide sequencing of MDS and AML specimens with monosomy 7 (-7) or 7q deletions [del(7q)] have revealed a remarkably low incidence of “second hit” mutations resulting in biallelic inactivation of any 7q gene with the exception of *EZH2,* which is only mutated in a small minority of cases.

*For example, whether EZH2 alterations have been linked to changes in OXPHOS gene expression was not addressed and could be briefly discussed. Given the finding about OXPHOS genes and ROS, ultrastructural analysis of the HSCs for changes in mitochondrial structure might be instructive but optional for this Short Report format*.

This is a good question as *EZH2* is a known myeloid TSG located at 7q36, which is distal to the 7q22*/5A3* CDS that we deleted (see revised Figure 1). Since many patients with MDS have monosomy 7 or large 7q deletions, it is possible that loss of *EZH2* and 7q22 genes cooperate to deregulate OXPHOS gene expression. We searched the literature, but did not find any studies linking *EZH2* mutations to abnormal OXPHOS gene expression. If the reviewers are aware of work addressing this question, we would be glad to cite it. We agree that pursuing the relationship of 7q deletions to OXPHOS gene expression is an interesting topic for further investigation, particularly given the transcriptional abnormalities in *5A3* HSC and evidence implicating OXPHOS in HSC homeostasis and aging. Interesting, two genes within the 7q22/5A3 deleted interval (*Pmpcb* and *Armc10),* encode proteins involved in mitochondrial quality control (see last paragraph of subsection “Changes in Gene Expression and Metabolic Activities in 5A3+/del Hematopoietic Cells”).

*The elevated CD150 hi population and the altered OxPhos expression raises the question of the state of quiescence of the HSCs. The manuscript notes unchanged BrdU incorporation (data not shown). It would be helpful to show a cell cycle analysis of the stem cell population if possible*.

Thank you for this suggestion. We added Figure 2—figure supplement 1 presenting these data. Specifically, we evaluated BrdU incorporation over a 24 hour time course, and also assessed cell division over 6 days by labeling cells with a N-hydroxysulfosuccinimide biotin derivative and quantitating the reduction in this label.